# The Effects of Subjective Social Class on Subjective Well-Being and Mental Health: A Moderated Mediation Model

**DOI:** 10.3390/ijerph20054200

**Published:** 2023-02-27

**Authors:** Kai Li, Feng Yu, Yanchi Zhang, Yongyu Guo

**Affiliations:** 1Department of Psychology, School of Philosophy, Wuhan University, Wuhan 430072, China; 2School of Psychology, Northwest Normal University, Lanzhou 730070, China; 3School of Psychology, Nanjing Normal University, Nanjing 210097, China

**Keywords:** subjective social class, self-class discrepancy, subjective social mobility, subjective well-being, mental health

## Abstract

In recent decades, China’s rapid economic growth has substantially improved average living standards; however, this has not been accompanied by greater happiness among the Chinese population. This phenomenon is known as the Easterlin Paradox (i.e., there is no link between a society’s economic development and its average level of happiness) in Western countries. This study examined the effects of subjective social class on subjective well-being and mental health in China. Consequently, we found that individuals in a relatively low social class had lower levels of subjective well-being and mental health; self-class discrepancy partially explains the relationship between subjective social class and subjective well-being and fully explains the relationship between subjective social class and mental health; and subjective social mobility moderates the path from self-class discrepancy to subjective well-being and mental health. These findings suggest that enhancing social mobility is an important method for reducing class differences in subjective well-being and mental health. These results have important implications, indicating that enhancing social mobility is an important method for reducing class differences in subjective well-being and mental health in China.

## 1. Introduction

Several studies have shown that social class is a powerful predictor of subjective well-being (SWB): the rich are happier than the poor [1,2,3,4], and the higher one’s status in the social hierarchy, the greater their happiness [5,6,7,8,9]. To the best of our knowledge, no study has yet examined the relationship between social class and SWB. Our study examined the role of self-class discrepancy and subjective social mobility in the relationship between social class, SWB, and mental health.

SWB refers to the manner in which individuals positively evaluate and experience their lives [10] and is often considered synonymous with the term “happiness” [4,11]. SWB consists of both cognitive and emotional components [10,12]. The cognitive component relates to how satisfied people are with their lives, both generally and in terms of specific domains (e.g., marriage, friends, and leisure), whereas the emotional component refers to how frequently people experience positive and negative affectivity. Thus, SWB is a measure of the quality of life [13] and conducive to psychological, physical, and interpersonal functioning [14].

Social class (or socioeconomic status) is defined as a hierarchical category based on one’s objective or subjective access to resources [15]. Objective social class is typically measured using a combination of indices related to educational attainment, annual income, and occupational status [15,16,17,18]. Subjective social class refers to how individuals perceive their own social class; that is, their self-perception of their standing in society relative to that of others [15,19].

Many studies have suggested that, in comparison to objective social class, subjective social class has a more stable and intense association with psychological, mental, and interpersonal functioning [20,21,22,23,24,25,26]. The social context in which individuals live is generally shaped by their perceptions of social rank [27]. Individuals who perceive themselves as having a high rank in society tend to have a strong sense of being able to predict, control, and influence their social environments without constraints [7]. In contrast, low-ranking individuals generally experience social constraints, helplessness, and uncertainty [28]. Thus, the latter believe themselves to be less valuable than others and less capable of affecting their social environment [27]. Therefore, relative rather than absolute income matters in this regard, implying that social comparison plays an important role in SWB [2]. Moreover, subjective social class can predict SWB and mental health even after accounting for objective measures of social class [20,22,24,29].

According to the multiple discrepancy theory, SWB is based on perceived discrepancies between what one has and what one wants, what others have, the best one has had in the past, what one expected to have three years ago, what one expects to have in five years, what one deserves, and what one needs. Among these perceived discrepancies, the discrepancy between what one has and wants is the most influential and strongest determinant of SWB [30]. Consequently, Michalos [30] suggested that discrepancies in relation to what one wants play a mediating role between all other discrepancies and SWB. Further, recent research has shown that multiple discrepancies are negatively associated with mental health and SWB. Specifically, those for whom these discrepancies are smaller experience more positive affectivity and less negative affectivity and have higher life satisfaction and better mental health [31].

Subjective social mobility refers to the extent to which people believe that they will enter a higher social class in the future [29,32,33]. Unlike actual social mobility, subjective social mobility focuses on people’s attitudes toward inequality and expectations of improving their social rank [29,34]. High social mobility improves life satisfaction, and subjective social mobility buffers the negative effects of income inequality on SWB [35]. Further, subjective social mobility mitigates the negative effect of subjective social class on SWB [29]. People with higher subjective social mobility usually believe that they have better opportunities to acquire a good education, earn money, and obtain prestigious jobs.

Considering this, we hypothesized that subjective social class is positively correlated with SWB (H1a) and mental health (H1b), subjective social class is negatively correlated with self-class discrepancy (i.e., the gap between one’s current situation and desired class) (H1c), self-class discrepancy negatively affects SWB (H2a) and mental health (H2b), and self-class discrepancy serves as a mediator between subjective social class and SWB and mental health (H3). Moreover, we posited that subjective social mobility moderates both the direct path from subjective social class to SWB and mental health (H4a) and the path from self-class discrepancy to SWB and mental health (H4b).

This study examined a moderated mediation model using a sample of college students in China. Specifically, in the model, we predicted that subjective social class would be positively correlated with SWB and mental health and that the relationship between subjective social class and SWB and mental health would be mediated by self-class discrepancy and moderated by subjective social mobility.

## 2. Materials and Methods

### 2.1. Participants

In order to obtain small–medium power (effect size *f*^2^ = 0.05 in a linear multiple regression analysis), a G*Power analysis suggested a total sample size of 159 participants was needed to obtain a power of 0.80 [36] (Faul et al. 2009). However, because we did not know the “true effect,” we oversampled in our study. Therefore, a total of 380 questionnaires were distributed in Wuhan, China, and all participants gave their informed consent. After excluding 24 invalid questionnaires (17 questionnaires were incomplete and 7 questionnaires were consistently answered using the same answer option), this study’s sample comprised 356 Han Chinese college students (281 female and 75 male), with a mean age of 20.32 (SD = 2.47). After completing anonymous questionnaires, each participant received a small amount of money in return for their participation.

### 2.2. Measures

#### 2.2.1. Subjective Social Class

We measured participants’ subjective social class using the MacArthur Scale [20]. Participants were shown a picture of a ladder with 10 rungs (1 = bottom rung and 10 = top rung), accompanied by the following description: “Think of the rungs of this ladder as representing where people stand in our society. At the top of the ladder are the people who are the best off —those who have the most income, the most education, and the best jobs. At the bottom are the people who are the worst off —those who have the lowest income, the least education, and the worst jobs or no jobs.” The participants were asked to consider their current situation and choose the rung that best represented where they believed they stood on the ladder. The results (M = 4.70, SD = 1.47) were then standardized.

#### 2.2.2. Self-Class Discrepancy

Next, participants were asked to imagine the social class they would like to attain and use sentences or phrases to describe the five most important characteristics of their desired social class. After completing the lists of characteristics, participants then indicated how closely they felt each of the characteristics they had listed described their current situation, using a five-point scale (1 = completely applies to my current situation, and 5 = does not apply to my current situation at all). The self-class discrepancy results were then averaged (M = 3.13, SD = 0.72) and standardized. Higher scores indicated that participants felt an extreme discrepancy between their current situation and desired social class. Cronbach’s alpha was 0.73 in this study.

#### 2.2.3. Subjective Social Mobility

We measured participants’ subjective social mobility using the Subjective Social Mobility Scale (SSMS) [29]. The SSMS consists of six items and assesses people’s belief that they will achieve a higher social status (e.g., “society currently allows me to improve my social status”). A five-point scale was used to measure responses (1 = strongly disagree and 5 = strongly agree). Item scores were then averaged (M = 3.13, SD = 0.53) and standardized. Cronbach’s alpha was 0.75 in this study.

#### 2.2.4. Subjective Well-Being

We measured participants’ SWB using the Satisfaction with Life Scale (SWLS) [37] and the Positive and Negative Affect Schedule (PANAS) [38]. The SWLS consists of five items that ask participants to provide a cognitive assessment of their life as a whole (e.g., “in most ways, my life is close to my ideal”). Both questionnaires were scored using a seven-point scale (1 = strongly disagree and 7 = strongly agree). The item scores were then averaged (M = 3.35, SD = 1.02). Cronbach’s alpha was 0.82 in this study.

The PANAS includes a positive affect scale (PAS) and a negative affect scale (NAS). Each subscale comprises 10 items that assess how frequently participants have experienced certain emotions (e.g., pride and distress) in recent weeks, with responses provided using a five-point scale (1 = very rarely or not at all, and 5 = extremely frequently). Item scores within each subscale were averaged (MPAS = 2.93, SDPAS = 0.59; MNAS = 3.46, SDNAS = 0.68) after reverse-coding the NAS items. Cronbach’s alphas for the PAS and NAS were 0.85 and 0.87, respectively. Next, the SWLS, PAS, and NAS scores were standardized and averaged to form an overall SWB index. Finally, the overall SWB index was standardized.

#### 2.2.5. Mental Health

We measured participants’ mental health using the short-form version of the Depression Anxiety Stress Scale (DASS21) [39]. The DASS21 consists of three seven-item subscales that assess whether participants have experienced depressive symptoms (e.g., “I felt that I had nothing to look forward to”), anxiety (e.g., “I felt scared without any good reason”), and/or stress (e.g., “I found it difficult to relax”) in recent weeks. The DASS21 is scored using a four-point scale (0 = did not apply to me at all, and 3 = applied to me very much or most of the time). Item scores within each subscale were averaged (MD = 2.39, SDD = 0.53; MA = 2.27, SDA = 0.47; MS = 2.04, SDS = 0.51) after all the items were reverse-coded. Next, the scores for depression, anxiety, and stress were standardized and averaged to form an overall mental health index. Finally, the overall mental health index was standardized. Cronbach’s alphas for depression, anxiety, and stress were 0.86, 0.78, and 0.80, respectively.

#### 2.2.6. Control Variables

We controlled for family income, the father’s education, and the mother’s education in the main analyses because of their potential associations with SWB and mental health.

### 2.3. Data Analyses

Gender was dummy-coded (0 = female, 1 = male), and all other variables were standardized. We then tested the associations between subjective social class, self-class discrepancy, subjective social mobility, SWB, and mental health. Next, we examined the model in which self-class discrepancy mediates the effects of subjective social class on SWB and mental health, and then we examined the model in which subjective social mobility moderates the direct effects of subjective social class on SWB and mental health and the indirect effects of subjective social class on SWB and mental health through self-class discrepancy (the path from self-class discrepancy to SWB and mental health).

We used IBM SPSS Version 19 to perform correlation analyses. For the two models, we used the PROCESS macro [40] to perform bias-corrected bootstrapping multiple mediation analyses with 5000 resamples. The mediating effects of self-class discrepancy were tested using Model 4, and the moderating role of subjective social mobility on these above effects was tested using Model 15.

## 3. Results

### 3.1. Correlation Analyses

First, we conducted a partial correlation analysis of the main variables, controlling for gender, age, family income, the father’s education, and the mother’s education. Table 1 presents the results. Consistent with H1, subjective social class was positively correlated with SWB (H1a) and mental health (H1b) and negatively correlated with self-class discrepancy (H1c). Self-class discrepancy was negatively correlated with SWB (H2a) and mental health (H2b). These results indicate that, relative to their lower-class counterparts, upper-class individuals have smaller discrepancies between their current situations and desired social class and are happier and mentally healthier. Further, self-class discrepancy appears to negatively affect happiness and mental health, while subjective social mobility promotes these traits.

### 3.2. Mediation Analyses

We used the PROCESS macro (Model 4, 5000 bootstrap resamples) [40] to examine whether self-class discrepancy mediates the relationship between subjective social class and SWB. The results are displayed in Figure 1A. Consequently, the indirect effect of subjective social class on SWB through self-class discrepancy was found to be significant (mindirect effect = 0.09, SE = 0.02, 95% CI = [0.05 to 0.14]), while the direct effect of subjective social class on SWB remained significant (*p* < 0.001). These results indicate a partial mediation effect.

We then used the PROCESS macro (Model 4, 5000 bootstrap resamples) to examine whether self-class discrepancy mediates the relationship between subjective social class and mental health. The results are shown in Figure 2A. Consequently, the indirect effect of subjective social class on mental health through self-class discrepancy was found to be significant (main direct effect = 0.04, SE = 0.02, 95%CI = [0.01 to 0.09]), while the direct effect of subjective social class on mental health was no longer significant after controlling for self-class discrepancy (*p* > 0.05). These results indicate that self-class discrepancy fully explains the effect of subjective social class on mental health. Thus, the results of the mediation analysis support H3.

### 3.3. Moderated Mediation Analyses

We used the PROCESS macro (Model 15, 5000 bootstrap resamples) to examine whether subjective social mobility moderates the first mediating model. The results are shown in Figure 1B. Here, the interaction between subjective social class and subjective social mobility was not significant (*p* > 0.05), suggesting that the relationship between subjective social class and SWB is not independent of subjective social mobility. However, the interaction between self-class discrepancy and subjective social mobility was significant (b2 = 0.16, SE = 0.04, 95% CI = [0.07 to 0.13]). Further, self-class discrepancy was found to be negatively associated with SWB in the low subjective social mobility (−SD) condition, β = −0.43, t = −6.40, *p* < 0.001, but not in the high subjective social mobility (+SD) condition, β = −0.11, t = −1.72, *p* > 0.05. In addition, subjective social mobility was positively associated with SWB, c2′ = 0.25, SE = 0.05, 95% CI = [0.16 to 0.35].

We then used the PROCESS macro (Model 15, 5000 bootstrap resamples) to examine whether subjective social mobility moderates the second mediating model. The results are shown in Figure 2B. Consequently, the interaction between subjective social class and subjective social mobility was not significant (*p* > 0.05), suggesting that the relationship between subjective social class and mental health is independent of subjective social mobility. However, the interaction between self-class discrepancy and subjective social mobility was significant (b2 = 0.14, SE = 0.05, 95% CI = [0.04 to 0.24]). Self-class discrepancy was negatively associated with mental health in the low subjective social mobility (−SD) condition, β = −0.27, t = −3.52, *p* < 0.001, but not in the high subjective social mobility (+SD) condition, β = 0.01, t = 0.12, *p* > 0.05. In addition, subjective social mobility was positively associated with SWB, c2′ = 0.11, SE = 0.06, 95% CI = [0.00 to 0.22].

## 4. Discussion

Social class is defined by both objective and subjective factors. Objectively, people in higher social classes have better access to material resources, such as income, education, and jobs [41,42]. Subjectively, social class emphasizes individuals’ perceived rank relative to others in society. For example, individuals in a high social class likely perceive that they have more money, a better education, and more prestigious jobs than others because they are of a higher social rank [25,43].

Several studies have suggested a link between people’s social class and their well-being and health [9,21,24,29,44]. Notably, compared with objective social class, subjective social class has been found to be more strongly associated with psychological and mental functioning [20,22,26,45]. Considering previous studies, we hypothesized that individuals who perceive themselves as being of a lower class will feel that they have a bigger self-class discrepancy and poorer SWB and mental health, and that self-class discrepancy mediates the relationship between subjective social class and SWB and mental health (a mediation model). In addition, we hypothesized that subjective social mobility moderates the mediation model.

Consistent with our hypotheses, the results showed that individuals who were subjectively in a lower social class experienced lower SWB and poorer mental health. Further, mediation analyses indicated that self-class discrepancies could partially explain the relationship between subjective social class and SWB and fully explain the relationship between subjective social class and mental health. Further, subjective social mobility was found to moderate the path from self-class discrepancy to SWB and mental health, while high subjective social mobility was found to weaken the association between self-class discrepancy and SWB and mental health. In addition, subjective social mobility was found to promote SWB and mental health.

### 4.1. Mediating Role of Self-Class Discrepancy

Self-class discrepancy refers to the gap between one’s current and desired social rank. Regarding the components of the self-class discrepancy, in addition to the ability to access material resources, such discrepancies may also include subjective feelings (e.g., sense of control, experience of relative deprivation) and social evaluations (e.g., prejudice and discrimination). When individuals self-report being part of a lower social class, they are actually stating that they have fewer resources and are subordinate to others. Such perceptions of having reduced resources and being subordinate can diminish one’s sense of personal control [15,19]. Abundant evidence supports the link between people’s sense of control and their SWB and mental health. People with a higher sense of control are likely to experience greater happiness and better mental health [31,46].

Another possible component of self-class discrepancy is relative deprivation. For example, people wish to achieve a social rank at which they can obtain what they feel they deserve. People have an innate ability to compare themselves to others and determine their social standing [47]. Downward comparisons increase well-being; however, upward comparisons have the opposite effect. In particular, when people believe that they have a relatively low social standing and that their low status is undeserved, they are likely to experience feelings of relative deprivation [48,49]. Studies have shown that feelings of relative deprivation diminish SWB and mental health [50,51]. Compared with those who are subjectively upper-class, subjectively lower-class individuals are more likely to support contextual explanations (e.g., political influence and discrimination) of economic inequality and broad social outcomes [19], as well as perceive a stronger sense of unfairness in society [15,52]. These factors all contribute to the feeling that one’s disadvantaged standing is undeserved [53,54,55].

In the present study, participants reported that obtaining respect was an important motivating factor in terms of their choice of desired social class. Several studies have shown that people in low social classes experience prejudice and discrimination [56,57]. For example, individuals from lower social classes are often stereotyped as lazy, incompetent, and unreliable [57,58,59]. These negative stereotypes can be used by lower-class people to justify their sense of inequality in society [57,60]. Notably, perceptions of prejudice and discrimination have been found to decrease people’s SWB and mental health [61,62,63].

### 4.2. Moderating Role of Subjective Social Mobility

Enhancing social mobility is an important method for reducing economic inequality [33], and social mobility can influence people’s attitudes and behaviors, such as their life satisfaction and fertility. Researchers believe that the most important variable in this regard is individuals’ subjective belief that they are moving up or down in society [32]. Thus, in this study, we examined the effects of subjective social mobility on SWB and mental health. Consequently, our results showed that subjective social mobility not only promotes SWB and mental health but also moderates the relationship between self-class discrepancy and SWB and mental health. Thus, the belief that one is improving one’s social class reduces the negative effect of self-class discrepancy on SWB and mental health.

One possible explanation for these findings is that having high subjective social mobility helps individuals believe that their social rank and those of others are fair and just. This belief may decrease people’s feelings of relative deprivation, and increase their personal sense of control. Subsequently, this belief strengthens the motivation to work harder and strive for a higher social status [33]. Although lower-class individuals currently have a greater self-class discrepancy, they believe that they can bridge the gap between their current and desired life situations and achieve high subjective social mobility through their own efforts. Therefore, high subjective social mobility can weaken the effects of self-class discrepancy on SWB and mental health.

### 4.3. Limitations and Future Directions

Some limitations of the present study should be noted. First, because of the correlational nature of the findings, causal interpretations are not appropriate. Although significant results were obtained, further experimental research should be conducted to confirm the causal relationships between the variables in question. Second, we considered some possible components of self-class discrepancy; however, we are not completely certain of the exact components that influence SWB and mental health. Consequently, the core components of self-class discrepancy that affect SWB and mental health still need to be identified. Third, the present results showed that subjective social mobility not only benefits SWB and mental health but also moderates the relationship between self-class discrepancy and SWB and mental health. However, we do not know the reason for this phenomenon. This issue should be addressed in future studies. Fourth, since the research samples were college students, we should recognize that the findings should be carefully used in a real-world setting. This limits the generalizability of our study, future studies should test the hypotheses with a broader range of participants, such as real employees or people of different ages and levels of education.

## 5. Conclusions

The present study found a significant and positive association between subjective social class and SWB and mental health and showed that self-class discrepancy partially mediates the relationship between subjective social class and SWB while fully mediating the relationship between subjective social class and SWB and mental health. In addition, subjective social mobility was found to play a moderating role in the path from self-class discrepancy to SWB and mental health. These findings suggest that enhancing social mobility is important for reducing class differences in SWB and mental health. In the practice of poverty alleviation, we should recognize that although inequality is profound and seen in all aspects of life, subjective social class has a big impact on SWB and mental health. Individuals who believe their system allows people to move up the social ladder will be more satisfied and less anxious. So, helping people to believe they can go from rags to riches is useful to increase their well-being and reduce their anxiety.

## Figures and Tables

**Figure 1 ijerph-20-04200-f001:**
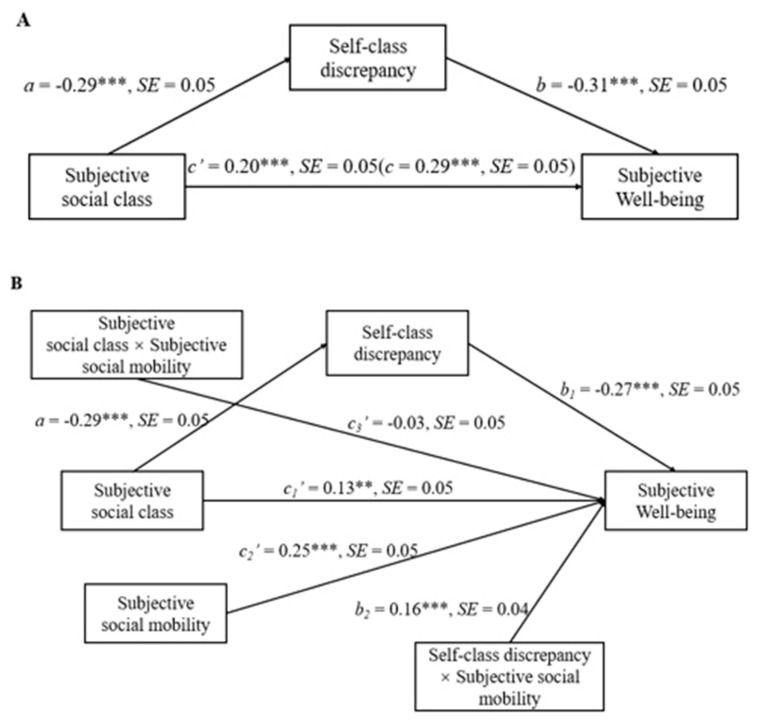
Model of the relationship between subjective social class and SWB mediated by self-class discrepancy (**A**), and moderated by subjective social mobility (**B**), controlling for gender, age, monthly family income, father’s education, and mother’s education. *** *p* < 0.001, ** *p* < 0.01.

**Figure 2 ijerph-20-04200-f002:**
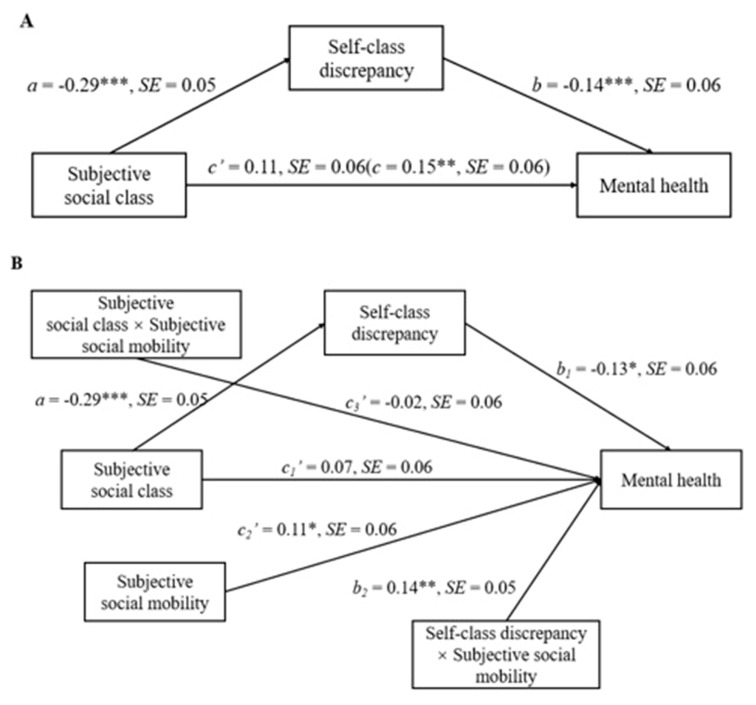
Model of the relationship between subjective social class and mental health mediated by self-class discrepancy (**A**), and moderated by subjective social mobility (**B**), controlling for gender, age, monthly family income, father’s education, and mother’s education. *** *p* < 0.001, ** *p* < 0.01, * *p* < 0.05.

**Table 1 ijerph-20-04200-t001:** Partial correlations between the main variables, controlling for gender, age, family income, father’s education, and mother’s education (*N* = 356).

Variables	1	2	3	4
(1) Subjective social class	—	—	—	—
(2) Self-class discrepancy	−0.277 ***	—	—	—
(3) Subjective social mobility	0.237 ***	−0.229 ***	—	—
(4) Subjective well-being	0.278 ***	−0.364 ***	0.341 ***	—
(5) Mental health	0.145 **	−0.172 **	0.160 **	0.572 ***

Note. ** *p* < 0.01, *** *p* < 0.001.

## Data Availability

Data will be provided by the authors on request.

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
