# Peer review of "The Effects of Subjective Social Class on Subjective Well-Being and Mental Health: A Moderated Mediation Model"

_ijerph, 2023, doi:10.3390/ijerph20054200_

Round 1
Reviewer 1 Report
Highly impressive work. No serious recommendations. Only some grammatical corrections:
Line 29 — Replace the semicolon with a colon and the period with a semicolon.
Line 123 — Period in place of the forward slash.
Line 181 — Remove ‘it’ or ‘self-class discrepancy’
Author Response
I have modified our manuscript according to your suggestions. I have replaced the semicolon with a colon and the period with a semicolon in Line 29, replaced the forward slash with period in Line 123, and remove ‘it’ in Line 181.
Thanks very much for these useful suggestions.
Reviewer 2 Report
Dear authors,
Thank you for your idea.
I have a few concerns;
Using a student sample is not appropriate as they will imagine the situation and give you their opinion as a student, not as real employees. this is a big limitation of this study.
You need to add a part about the sample and population and how can you use the findings in a real setting.
You need to add a part about the sample and population and how can you use the findings in a real setting.
Finally, you need to add more details to the conclusion part, adding another part to the practical implications.
Good luck
Author Response
Thanks very much for these useful suggestions.
We added limitations in limitations and future directions.
Forth, the research samples were college students, we should recognize that the findings should be careful be used in real setting. This limits the generalizability of our study, future studies should test the hypotheses with a broader range of participants, such as real employees or people of different ages, levels of education.
We adding a small part to the practical implications.
These findings suggest that enhancing social mobility is important for reducing class differences in SWB and mental health. Individuals who believe their system allow people to move up the social ladder will be more satisfied and less anxious. Helping people to believe they can from rags to riches is useful to increase their well-being.
Reviewer 3 Report
Did you describe limitations? Use of students limits the generalizability of the study. This should be acknowledged.
Author Response
I have modified our manuscript according to your suggestions. Thanks very much for your useful suggestions.
Forth, the research samples were college students, we should recognize that the findings should be careful be used in real setting. This limits the generalizability of our study, future studies should test the hypotheses with a broader range of participants, such as real employees or people of different ages, levels of education.